# Systematic Review, Meta-Analysis and Radiomics Quality Score Assessment of CT Radiomics-Based Models Predicting Tumor EGFR Mutation Status in Patients with Non-Small-Cell Lung Cancer

**DOI:** 10.3390/ijms241411433

**Published:** 2023-07-14

**Authors:** Mehdi Felfli, Yan Liu, Fadila Zerka, Charles Voyton, Alexandre Thinnes, Sebastien Jacques, Antoine Iannessi, Sylvain Bodard

**Affiliations:** 1Median Technologies, F-06560 Valbonne, France; mehdi.felfli@mediantechnologies.com (M.F.); yan.liu@mediantechnologies.com (Y.L.); fadila.zerka@mediantechnologies.com (F.Z.); charles.voyton@mediantechnologies.com (C.V.); alexandre.thinnes@mediantechnologies.com (A.T.); sebastien.jacques@mediantechnologies.com (S.J.); antoine.iannessi@mediantechnologies.com (A.I.); 2Centre Antoine Lacassagne, F-06100 Nice, France; 3AP-HP, Service d’Imagerie Adulte, Hôpital Necker Enfants Malades, Université de Paris Cité, F-75015 Paris, France; 4CNRS UMR 7371, INSERM U 1146, Laboratoire d’Imagerie Biomédicale, Sorbonne Université, F-75006 Paris, France

**Keywords:** radiomics quality score assessment, CT radiomics-based models, tumor EGFR mutation status, non-small-cell lung cancer

## Abstract

Assessment of the quality and current performance of computed tomography (CT) radiomics-based models in predicting epidermal growth factor receptor (EGFR) mutation status in patients with non-small-cell lung carcinoma (NSCLC). Two medical literature databases were systematically searched, and articles presenting original studies on CT radiomics-based models for predicting EGFR mutation status were retrieved. Forest plots and related statistical tests were performed to summarize the model performance and inter-study heterogeneity. The methodological quality of the selected studies was assessed via the Radiomics Quality Score (RQS). The performance of the models was evaluated using the area under the curve (ROC AUC). The range of the Risk RQS across the selected articles varied from 11 to 24, indicating a notable heterogeneity in the quality and methodology of the included studies. The average score was 15.25, which accounted for 42.34% of the maximum possible score. The pooled Area Under the Curve (AUC) value was 0.801, indicating the accuracy of CT radiomics-based models in predicting the EGFR mutation status. CT radiomics-based models show promising results as non-invasive alternatives for predicting EGFR mutation status in NSCLC patients. However, the quality of the studies using CT radiomics-based models varies widely, and further harmonization and prospective validation are needed before the generalization of these models.

## 1. Introduction

Targeted therapy for non-small-cell lung carcinoma (NSCLC) is rapidly becoming a standard treatment approach in oncology [1]. The identification of molecular factors in lung cancers is crucial for selecting appropriate targeted therapies. Particular interest was shown towards mutations of the epidermal growth factor receptor (EGFR) due to their presence in important proportions of patients with lung adenocarcinomas (up to 50% in Asian populations and 10–35% in Western populations), as well as their ability to predict therapeutic response to EGFR-specific tyrosine kinase inhibitors (TKIs) [2,3,4]. Current methods for detecting EGFR mutations involve obtaining tissue specimens invasively through surgery or needle biopsy. These methods are costly and time-consuming and lack spatial resolution to describe intra-tumor heterogeneity and the complexity of the tumor microenvironment [5]. On the other hand, liquid biopsies are non-invasive alternatives to traditional tissue biopsies and can provide valuable information. However, their limitations include lower sensitivity due to low tumor-derived material, the possibility of detecting mutations not associated with the tumor, and the cost and availability of tests [6].

Radiomics-based models using computed tomography (CT) scans offer a promising non-invasive alternative for predicting EGFR mutation status in NSCLC patients [7]. By extracting quantifiable features from medical images obtained via radiomics methods and using them to train machine learning models, CT radiomics-based models can generate a signature for the molecular status of tumors, thus reflecting the abundant information and multi-dimensionality of the entire tumor and its surrounding tissue. Radiomics workflows involve several general steps, such as image acquisition, segmentation, feature extraction, and model building. These can be limited by factors such as image quality, data preprocessing, and model validation [7,8]. The number of new publications on radiomics has been increasing. However, the techniques and software used for feature extraction vary from one study to another, and most studies include small patient cohorts with a high number of extracted features, lacking validation [9,10,11,12]. Therefore, a systematic literature review was deemed necessary to evaluate the accurate and actual performance of CT radiomics-based models for predicting EGFR mutation status in NSCLC patients.

Overall, this systematic review aims to assess the quality and present effectiveness of CT radiomics-based models in predicting EGFR mutation status using a systematic approach.

## 2. Materials and Methods

This systematic review was conducted according to the Preferred Reporting Items for Systematic Reviews and Meta-analysis for Diagnostic Test Accuracy (PRISMA-DTA) statement [13]. This statement describes an evidence-based minimum set of items for reporting in systematic reviews and meta-analyses of diagnostic studies and is registered under the number CRD42023411128 (PROSPERO).

### 2.1. Search Strategy

A systematic search for eligible publications published up to 27 April 2023 was performed in the PubMed and Google Scholar databases using the keywords “Radiomics”, ”Model”, “CT scan”, “NSCLC”, “Lung”, and “EGFR”. Only original studies were selected.

The PubMed and Google Scholar databases were chosen for the references’ high quality and ease of accessibility. Two reviewers assessed the eligibility of the articles identified through the search: the first reviewer screened each record, each report, as well as the reference lists of the included studies; the second reviewer checked the results by following the same process.

### 2.2. Inclusion and Exclusion Criteria

Only publications on original research studies were considered for inclusion. Studies were deemed relevant for inclusion if they included cohorts of pathology-confirmed NSCLC patients who had CT scans before surgery or biopsy, did not receive anti-tumor treatment before the CT scan, and had gene testing for EGFR mutation. The analysis of the CT radiomics or CT radiomics-based model had to be conducted to predict the EGFR mutation status. Models trained on deep learning features were not considered to focus on radiomics features.

Excluded published articles included reviews, meta-analyses, case reports, conference abstracts, letters to the editor, comments, posters, technical reports, duplicate studies, publications not in English, and non-human studies. Moreover, original research articles were excluded if the model’s performance was not assessed via the area under the receiver operating characteristic curve (ROC AUC) or indirect/combined measures.

The process for identifying and selecting published articles is further detailed in Figure 1.

### 2.3. Data Extraction

The following data were extracted upon availability: name of the first author, year of publication, used standard reference, sample size, cohort origin, histological subtype, feature extraction, feature reduction and modeling methods, number of involved centers (either single or multicenter), study objectives, and prediction performance in terms of ROC AUC with its corresponding confidence interval (CI). Data extraction was performed independently by two reviewers. As a quality control measure, extracted data were compared, and consensus resolved disagreements.

### 2.4. Quality Analysis

Radiomics is a multi-step process with methodological challenges to ensure the findings’ robustness, reproducibility, and generalizability. Therefore, studies reporting correlations between imaging biomarkers and patient outcomes need standardized acquisition and processing techniques to confirm causality. To improve the radiomics workflow, a set of recommendations was proposed in 2017 by Lambin et al. [7], including the Radiomics Quality Score (RQS). This tool has become the standard for systematic literature reviews on radiomics quality assessment [14].

The RQS includes 16 items that assess protocol quality, reproducibility, reporting, validation, biological/clinical validation and utility, performance index, demonstration of a higher evidence level, and open science. The total scale score ranges from −8 to +36; higher scores indicate better quality, and a percentage score of 100% corresponds to a score of +36 [7].

### 2.5. Statistical Analysis

Statistical analysis was conducted using the Python 3.7 software. Forest plots were generated using the MedCalc Statistical Software (version 19.2.6; MedCalc Software bv, Ostend, Belgium; https://www.medcalc.org (accessed on 25 May 2023)). The RQS scores and percentages were summed and calculated as described by Lambin et al. [7].

Various measures were employed to assess the risk of bias in the included studies. The z-score and heterogeneity test I^2^ value were used to evaluate the heterogeneity across the studies. Funnel plots were generated to assess publication bias, and Egger’s test and Kendall’s tau value were used to quantify funnel plot asymmetry.

## 3. Results

A total of 140 full-text manuscripts were identified using the pre-specified search terms. After excluding 106 publications (20 duplicated publications and 86 articles meeting at least one of the exclusion criteria), 28 were included [3,9,10,11,12,15,16,17,18,19,20,21,22,23,24,25,26,27,28,29,30,31,32,33,34,35,36,37]. The retained publications targeted the prediction of the EGFR mutation status using radiomics and/or an AI-based radiomics model, with ROC AUCs as a performance metric. Articles on studies evaluating the EGFR as secondary endpoints relative to survival or therapeutic response were excluded.

The quality of the 28 included publications was assessed using the RQS tool. Scores ranged from 11 (30.56%) to 24 (66.67%). The mean score obtained with the RQS was 15.25, accounting for 42.34% of the total score. The median score was 41.67% (interquartile range [IQR] = 15). The RQS of each publication included in this literature review is presented in Figure 2.

All studies used gold standards using radiologist segmentation for lesions and Polymerase Chain reaction (PCR) testing or sequencing for EGFR mutation status. The feature selection steps were performed in a total of 26 studies. However, none were prospective or reported phantom studies and cost-effectiveness analyses. Imaging feature extraction at multiple time points was performed in one study only [34].

Other items assessed by the RQS, including “Validation”, “Cut-off analyses”, “Calibration statistics”, “Multivariable analysis”, and “Open science and data” varied the most between studies. More specifically, resampling method techniques were widely applied (24/28 studies). However, calibration statistics and their statistical significance were performed in fewer studies (8/28 studies), and the determination of risk groups was performed in a total of 4/28 studies. An overview of the RQS item scores is provided in Figure 3, and the RQS of each included article (overall and item scores [7]) is provided in Appendix A.

Regarding radiomics workflow, most studies used the Pyradiomics or LifeX feature extraction tools. However, one study (Lu et al., 2020) utilized the Imaging Biomarker Explorer (IBEX) tool [27], and others used an in-house tool. The extracted features varied between the studies, ranging from 11 to 4306 (median number: 580), with an average of 751 ± 866 features. The most commonly used machine learning models for classification were logistic regression, random forest, and support vector machine. All the studies were retrospective; five used data from The Cancer Imaging Archive (TCIA) open-source repository. For ground truth, delineation performed by radiologists was used for segmentation, and PCR or sequencing was used to determine the EGFR mutation status. Correlation coefficients and least absolute shrinkage and selection operator (LASSO) were the two most commonly used methods for feature reduction.

Based on the forest plot analysis, which included 14 studies from 28 that reported ROC AUC confidence intervals, the pooled ROC AUC value for the performance of CT radiomics-based models for the prediction of the EGFR mutation status in NSCLC patients was 0.8, with a 95% CI of [0.757–0.845] indicating moderate-to-high prediction accuracy (Figure 4).

The z-score was 35.459 with a *p*-value of less than 0.001, indicating that the pooled ROC AUC value was statistically significant (Table 1).

However, the studies had high heterogeneity, with an I-squared value of 82.78% and a *p*-value of less than 0.0001. This heterogeneity may be attributed to differences in the included patient populations, study designs, manufacturers, and protocols, as well as the used model and radiomics features.

Egger’s test resulted in an intercept of −0.013, indicating that there may have been a small degree of asymmetry in the funnel plot (Figure 5).

However, this asymmetry was not statistically significant, suggesting no evidence of publication bias in this meta-analysis. The Kendall’s tau value was 0.098, indicating a weak positive correlation between the effect size and its standard error. This could have been due to the publication bias, relative to studies with a small database and low performance not being published, or to other sources of heterogeneity related to variations in radiomics workflows.

## 4. Discussion

The quality of the studies included in this meta-analysis was assessed using the RQS tool. It should be noted that this tool did not provide a definitive assessment of the study quality but rather identified potential methodological weaknesses in radiomics studies. Also, the RQS rating was subject to subjective assessments and could vary between raters. Measurement of the inter-rater variability was not performed in this meta-analysis.

The RQS results obtained in this meta-analysis also revealed heterogeneity in the qualities and methodologies of the included studies, primarily due to the number of steps in the radiomics workflow. This indicated the need for improved methodological rigor in the conduct and reporting of future radiomics studies in order to increase the reliability and reproducibility of CT radiomics-based models for predicting EGFR mutation status.

This meta-analysis of studies investigating the performance of CT radiomics-based models for predicting EGFR mutation status in NSCLC patients yielded several significant findings. First, the analysis demonstrated that radiomics analysis and radiomics models could provide accurate predictions of the EGFR mutation status using CT imaging data. The analysis of the ROC AUC suggested that these models had a high degree of diagnostic accuracy with an overall ROC AUC of 0.8 (95% CI: 0.757–0.845), which agreed with previous publications [38]. Radiomics-based models performed significantly better than clinical features, as demonstrated by multivariable analyses with non-radiomics features [3,10,16,19,20,24,27,30,31,32,35,36,37].

Furthermore, a combined model with clinical and radiomics features seemed to have better results than only radiomics-based models alone [9,16,37].

The performed analyses also suggested that radiomics-based analyses combined with radiologist assessment could be helpful in clinical decision-making for NSCLC patients. By helping the accurate prediction of the EGFR mutation status, CT radiomics-based methods can help clinicians identify NSCLC patients who may benefit from targeted therapies, improving treatment outcomes and reducing morbidity and mortality.

Noninvasive CT radiomics-based models offer distinct advantages over traditional methods. These models utilize existing medical imaging data, eliminating the need for invasive procedures and reducing patient discomfort, risk, and potential complications [39]. Indeed, CT scans are a routine diagnostic procedure, providing readily available data for radiomics-based models without requiring additional invasive procedures or specialized testing. Unlike current methods that sample only a small portion of the tumor, CT radiomics-based models can analyze the entire tumor volume with tumor microenvironment, offering a more comprehensive evaluation of tumor characteristics [5,7,40]. These models employ advanced imaging analysis techniques to extract quantitative features, objectively assessing tumor characteristics such as shape, texture, and density. Quantifying these features could enhance the accuracy and reliability of predicting genetic mutations [8,41]. Moreover, serial CT scans enable longitudinal monitoring of tumor characteristics, facilitating assessment of treatment response, disease progression evaluation, and monitoring emerging genetic mutations over time. It is important to note that CT radiomics-based models should not replace confirmatory molecular testing methods but can serve as valuable adjunct tools to assist clinicians in decision-making, risk stratification, and treatment planning.

This meta-analysis also revealed limitations and gaps in the current literature. For example, while radiomics-based models have shown promising results in predicting EGFR mutation status, the studies included in the current meta-analysis were largely retrospective, and some were conducted on a small sample size (less than 60 cases) [11,28,33]. Also, most of the studies lacked external validation and multicentric data sources. Future studies should aim to prospectively validate these models on more extensive and more diverse patient populations, as well as investigate their performance in real-world clinical settings. Given the high degree of heterogeneity, harmonization and reproducibility remain the main obstacles to generalizing CT radiomics-based methods [42,43].

Overall, this meta-analysis suggests that CT radiomics-based models have the potential to be a valuable tool in predicting EGFR mutation status in NSCLC patients, but further research is needed to fully establish their clinical utility and address the limitations and gaps in the current literature.

## 5. Conclusions

In conclusion, despite the heterogeneity and methodological weaknesses observed in the methodology of studies, this meta-analysis highlights that CT radiomics-based models show promise in accurately predicting EGFR mutation status in NSCLC patients with high diagnostic accuracy. These noninvasive techniques offer advantages over invasive procedures, could comprehensively evaluate tumor characteristics, and might enable longitudinal monitoring. However, the current literature lacks external validation, multicentric data sources, and prospective studies on larger and more diverse patient populations. Harmonization and reproducibility remain crucial challenges.

## Figures and Tables

**Figure 1 ijms-24-11433-f001:**
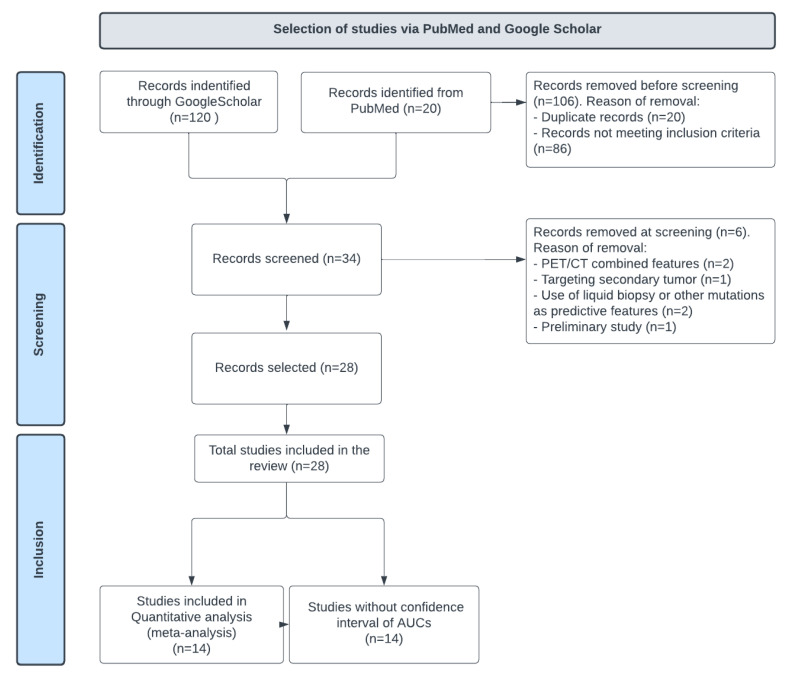
Selection process pipeline.

**Figure 2 ijms-24-11433-f002:**
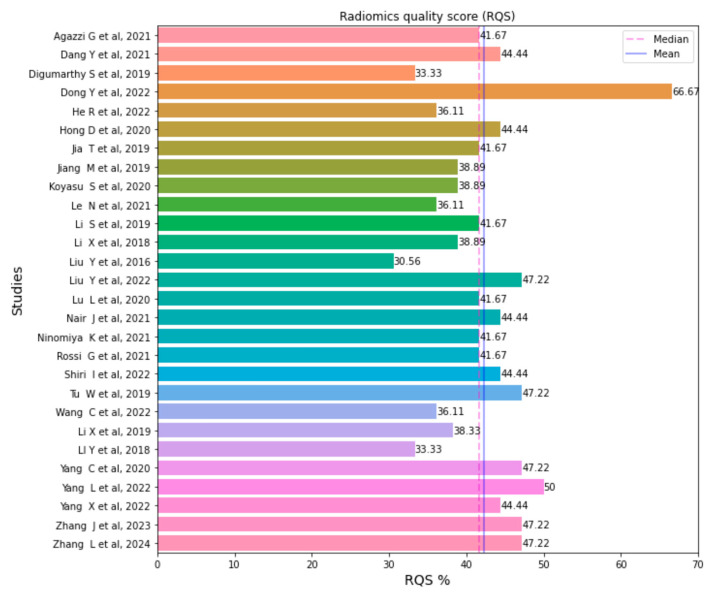
Bar plot of attributed radiomics quality score (RQS) by study paper [3,9,10,11,12,15,16,17,18,19,20,21,22,23,24,25,26,27,28,29,30,31,32,33,34,35,36,37].

**Figure 3 ijms-24-11433-f003:**
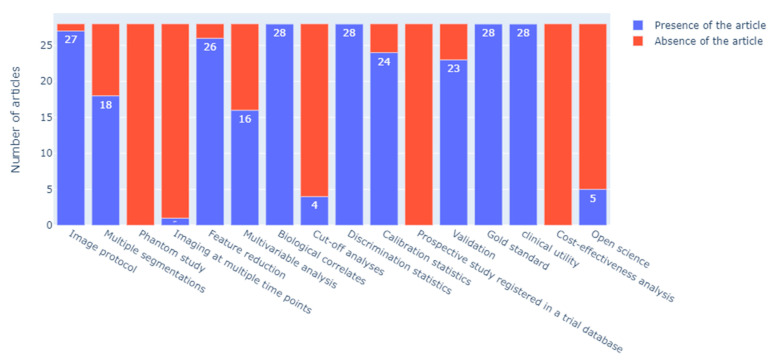
RQS item scores of the studies included in this meta-analysis (the sum reported is the number of studies with at least one point in the RQS workflow step).

**Figure 4 ijms-24-11433-f004:**
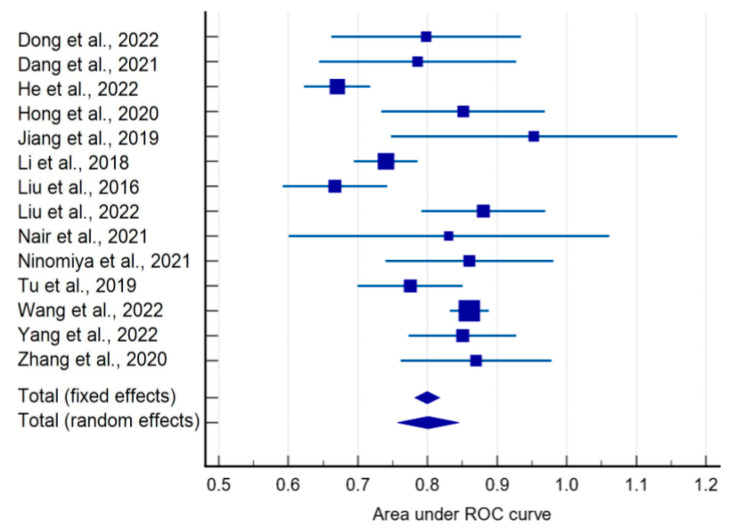
Forest plot of ROC AUCs [3,9,10,11,12,17,18,19,24,25,26,28,31,36].

**Figure 5 ijms-24-11433-f005:**
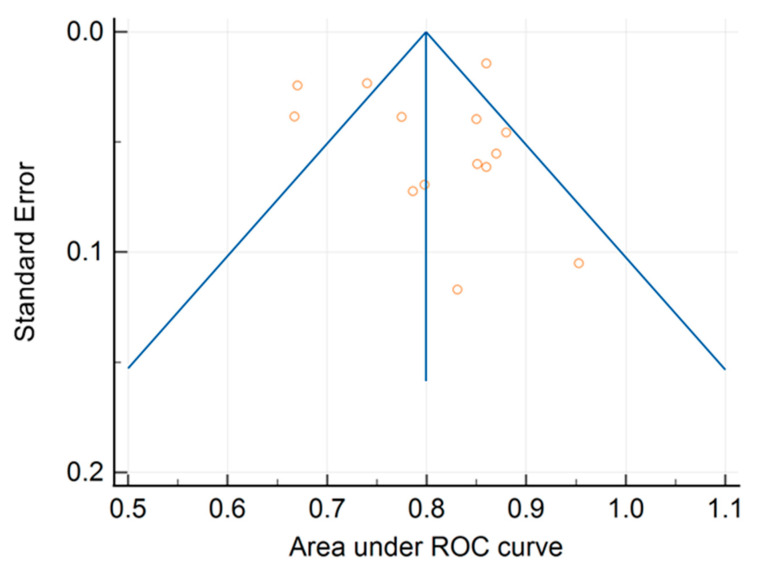
Funnel plot of ROC AUCs with 95% confidence limits.

**Table 1 ijms-24-11433-t001:** Forest plot statistics.

Study	ROC Curve Area	Standard Error	95% CI	z-Score	*p*-Value	Weight	(%)	Sample Size	Study Centers
						Fixed	Random		
Dong et al., 2022 [17]	0.798	0.0694	0.662 to 0.934			1.68	5.47	132	Single center
Dang et al., 2021 [3]	0.786	0.0723	0.644 to 0.928			1.55	5.24	118	Multicenter
He et al., 2022 [18]	0.670	0.0243	0.622 to 0.718			13.67	9.99	758	Multicenter
Hong et al., 2020 [19]	0.851	0.0600	0.733 to 0.969			2.25	6.29	449	Single center
Jiang et al., 2019 [12]	0.953	0.105	0.747 to 1.000			0.73	3.28	80	Single center
Li et al., 2018 [24]	0.740	0.0233	0.694 to 0.786			14.91	10.09	810	Single center
Liu et al., 2016 [25]	0.667	0.0384	0.592 to 0.742			5.50	8.52	397	Single center
Liu et al., 2022 [26]	0.880	0.0456	0.791 to 0.969			3.90	7.74	211	Single center
Nair et al., 2021 [11]	0.831	0.117	0.600 to 1.000			0.59	2.79	80	Single center
Ninomiya et al., 2021 [28]	0.860	0.0614	0.740 to 0.980			2.15	6.16	194	Multicenter
Tu et al., 2019 [31]	0.775	0.0386	0.699 to 0.851			5.45	8.50	404	Single center
Wang et al., 2022 [9]	0.860	0.0143	0.832 to 0.888			39.82	10.81	3629	Single center
Yang et al., 2022 [10]	0.850	0.0396	0.772 to 0.928			5.16	8.39	1028	Single center
Zhang et al., 2020 [36]	0.870	0.0552	0.762 to 0.978			2.65	6.74	728	Single center
Total (fixed effects)	0.800	0.00900	0.782 to 0.817	88.858	<0.001	100.00	100.00		
Total (random effects)	0.801	0.0226	0.757 to 0.845	35.459	<0.001	100.00	100.00		

## Data Availability

Not applicable.

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
