# Peer review of "Systematic Review, Meta-Analysis and Radiomics Quality Score Assessment of CT Radiomics-Based Models Predicting Tumor EGFR Mutation Status in Patients with Non-Small-Cell Lung Cancer"

_ijms, 2023, doi:10.3390/ijms241411433_

Round 1

Reviewer 1 Report

The review manuscript by Felfli et al is an interesting study evaluating a noninvasive model for predicting tumor EGFR mutation status in patients with non-small-cell lung cancer. I only have a few minor suggestions for the authors:

1. In my understanding, the title of the manuscript has the scope of minor improvement, such as: "Systematic Review and Meta-Analysis of Radiomics Quality Score Assessment of CT Radiomics-Based Models Predicting Tumor EGFR Mutation Status in Patients with Non-Small-Cell Lung Cancer". 

2. Line 147: "28 manuscripts were included" should be written as '28 articles were included' given that these were published reports.

3. Lines 161 and 186: Authors mentioned sequencing was used but did not provide the details of type of sequencing used- Sanger or NGS or both? 

4. Discussion can be elaborated. Please discuss in detail how this noninvasive model is superior or more reliable predictor over other traditional methods? 

5. Conclusions have no information. Please rewrite conclusions with a clear message that was synthesized in this study. 

Overall, English language used is fine. 

Author Response

Point-by-point response to the reviewer’s comments

Dear Referees,

We would like to thank the reviewers and the editorial board for their constructive feedbacks. We modified the text in order to address their criticism. We believe the article was improved and made more precise in the process. The editor and reviewers will find below our response for each of the points they raised.

Reviewer 1

The review manuscript by Felfli et al is an interesting study evaluating a noninvasive model for predicting tumor EGFR mutation status in patients with non-small-cell lung cancer. I only have a few minor suggestions for the authors:

Answer: Thank you very much for your comment. We have made the modifications.

  1. In my understanding, the title of the manuscript has the scope of minor improvement, such as: "Systematic Review andMeta-Analysis ofRadiomics Quality Score Assessment of CT Radiomics-Based Models Predicting Tumor EGFR Mutation Status in Patients with Non-Small-Cell Lung Cancer". 

Answer: Absolutely. We made the correction.

  1. Line 147: "28 manuscripts were included" should be written as '28 articles were included' given that these were published reports.

Answer: Absolutely. We made the correction.

  1. Lines 161 and 186: Authors mentioned sequencing was used but did not provide the details of type of sequencing used- Sanger or NGS or both? 

Answer : unfortunately this information is not specified in the studies in question.

  1. Discussion can be elaborated. Please discuss in detail how this noninvasive model is superior or more reliable predictor over other traditional methods? 

 Answer: we completed this part:

“ (…) Noninvasive CT radiomics-based models offer distinct advantages over traditional methods. These models utilize existing medical imaging data, eliminating the need for invasive procedures and reducing patient discomfort, risk, and potential complications [39]. Indeed, CT scans are a routine diagnostic procedure, providing readily available data for radiomics-based models without requiring additional invasive procedures or specialized testing. Unlike current methods that sample only a small portion of the tumor, CT radiomics-based models can analyze the entire tumor volume with tumor microenvironment, offering a more comprehensive evaluation of tumor characteristics [5, 7, 40]. These models employ advanced imaging analysis techniques to extract quantitative features, objectively assessing tumor characteristics such as shape, texture, and density. Quantifying these features could enhance the accuracy and reliability of predicting genetic mutations [8, 41]. Moreover, serial CT scans enable longitudinal monitoring of tumor characteristics, facilitating assessment of treatment response, disease progression evaluation, and monitoring emerging genetic mutations over time. It is important to note that CT radiomics-based models should not replace confirmatory molecular testing methods but can serve as valuable adjunct tools to assist clinicians in decision-making, risk stratification, and treatment planning. (…)”

  1. Conclusions have no information. Please rewrite conclusions with a clear message that was synthesized in this study. 

Answer: Absolutely. We modified the conclusion taking into account the comments of the two reviewers :

“In conclusion, despite the heterogeneity and methodological weaknesses observed in the methodology of studies, this meta-analysis highlights that CT radiomics-based models show promise in accurately predicting EGFR mutation status in NSCLC patients with high diagnostic accuracy. These noninvasive techniques offer advantages over invasive procedures, could comprehensively evaluate tumor characteristics, and might enable longitudinal monitoring. However, the current literature lacks external validation, multicentric data sources, and prospective studies on larger and more diverse patient populations. Harmonization and reproducibility remain crucial challenges.  “

Reviewer 2 Report

This article on Meta-Analysis, and Radiomics Quality 2 Score Assessment seems well-written, organized, and worth pursuing.

 I think this is acceptable. Suggestions are merely methodological:

- I don't see the role of the first table, therefore I think it could be inserted in the text;

- To reword the text in lines 121, 137 and 179, leaving only the citations leading to the references;

- The conclusions section is well-written but not exciting.

Minor editing of English language required.

Author Response

Point-by-point response to the reviewer’s comments

Dear Referees,

We would like to thank the reviewers and the editorial board for their constructive feedbacks. We modified the text in order to address their criticism. We believe the article was improved and made more precise in the process. The editor and reviewers will find below our response for each of the points they raised.

(Reviewer 2)

This article on Meta-Analysis, and Radiomics Quality 2 Score Assessment seems well-written, organized, and worth pursuing.

 I think this is acceptable. Suggestions are merely methodological:

Answer: Thank you very much for your comment. We have made the modifications.

- I don't see the role of the first table, therefore I think it could be inserted in the text;

Answer: Absolutely. We made the correction.

- To reword the text in lines 121, 137 and 179, leaving only the citations leading to the references;

 Answer: Absolutely. We made the modifications.

- The conclusions section is well-written but not exciting.

Answer: Absolutely. We modified the conclusion taking into account the comments of the two reviewers :

“In conclusion, despite the heterogeneity and methodological weaknesses observed in the methodology of studies, this meta-analysis highlights that CT radiomics-based models show promise in accurately predicting EGFR mutation status in NSCLC patients with high diagnostic accuracy. These noninvasive techniques offer advantages over invasive procedures, could comprehensively evaluate tumor characteristics, and might enable longitudinal monitoring. However, the current literature lacks external validation, multicentric data sources, and prospective studies on larger and more diverse patient populations. Harmonization and reproducibility remain crucial challenges.  “
